# HYBRID RETRIEVAL-AUGMENTED GENERATION FOR REAL-TIME COMPOSITION ASSISTANCE

## ABSTRACT

Retrieval augmentation enhances performance of traditional language models by incorporating additional context. However, the computational demands for retrieval augmented large language models (LLMs) pose a challenge when applying them to real-time tasks, such as composition assistance. To address this limitation, we propose the Hybrid Retrieval-Augmented Generation (HybridRAG) framework, a novel approach that efficiently combines a cloud-based LLM with a smaller, client-side, language model through retrieval augmented memory. This integration enables the client model to generate effective responses, benefiting from the LLM's capabilities and contextual information. Additionally, through an asynchronous memory update mechanism, the client model can deliver real-time completions swiftly to user inputs without the need to wait for responses from the cloud. Our experiments on the Wikitext dataset and Pile subsets demonstrate that HybridRAG significantly improves utility over client-only models while maintaining low latency.

## 1 INTRODUCTION

Retrieval-augmented approaches (Lewis et al., 2020; Liu et al., 2022) have emerged as a powerful tool to boost Large Language Model (LLM) performance by incorporating external documents (Lewis et al., 2020; Liu et al., 2022). This integration enables models such as GPT-3 (Brown et al., 2020) and ChatGPT to leverage external contextual information, resulting in improved contextual understanding, streamlined integration of private data, and reduced occurrence of hallucinations. However, the retrieval-augmented large language models can be slow and expensive to run due to the size of the model and the extra retrieval step they require, which can cause latency and limit its application in tasks requiring real-time responses, such as composition assistance.

Real-time composition tools are designed to swiftly suggest next words or sentences, and therefore operate within tight latency budgets (typically in the order of 100ms or less). To avoid latency overheads for sending inference requests to the cloud, these models are usually deployed on users' edge devices. This imposes strict constraints on the model's size and capabilities, limiting the effectiveness of composition assistance. While recent advancements have enabled LLMs like LLAMA (Touvron et al., 2023) to generate 5 tokens per second on a smartphone[1], they still fall short in terms of achieving real-time response time for completing a sentence within a few hundred milliseconds. In addition, embedding a retrieval-augmentation module into the edge model may not always be ideal because relevant documents are often stored in the cloud, such as in an organization's centralized database, and the retrieval step can introduce additional latency overhead.

To address these challenges, we propose the Hybrid Retrieval-Augmented Generation (HybridRAG) framework. This framework leverages cloud-generated memory augmentation to boost the performance of small language models on edge devices, while operating *in an asynchronous manner*. The HybridRAG framework consists of a retriever model and memory generator residing on the cloud server, as well as an augmentation coordinator and memory-augmented client model deployed on client devices. The cloud model creates the retrieval-augmented memory and sends it asynchronously to the client model. This allows the client model to respond to user requests for suggestions in real-time without waiting for the cloud memory. This asynchronous communication

---

[1] https://news.ycombinator.com/item?id=35171116

also reduces the computational cost of the cloud model, as it does not need to process every new input from the user.

In summary, the contributions in this work can be summarized as follows:

- *Hybrid retrieval-augmentation enables real-time generation:* A novel hybrid framework is proposed to enable real-time text generation on client devices, utilizing retrieval augmentation in the cloud server. Our approach leverages asynchronous client-cloud communication to achieve fast responses while mitigating the effects of network latency and avoiding slow inference inherent to cloud-based retrieval-augmented LLMs.

- *Enhanced utility:* We introduce a LLM-augmented memory approach to enhance the utility of the client language model, using LLM–generated labels for instruction-tuning of the client model. Our model effectively utilizes the LLM-augmented memory, resulting in substantial improvement in client model performance.

- *Reduced client-to-cloud communication:* Our augmentation coordinator module enables asynchronous memory augmentation, minimizing the client-to-cloud communication by requesting augmented memory only when existing memory becomes stale. Additionally, utilizing LLM-compressed memory further minimizes data transfer volume.

To evaluate the efficacy of our proposed method, we conducted experiments on five benchmark datasets from diverse domains. Our model outperformed the top-performing hybrid baseline, achieving a notable average improvement of 48.6% in GLEU score through the use of LLM-generated memory augmentation, and a further 9.5% improvement through instruction tuning on the client model. In addition, our asynchronous framework demonstrated substantial speed improvement compared to a synchronous approach under the same experimental setup. We plan to make our code and data public upon publication of the paper.

## 2 RELATED WORK

**Hybrid Computing**  Hybrid computing between edge and cloud devices originated outside the realm of machine learning. It typically divides processing tasks between the edge and the cloud, effectively addressing the limited computation capabilities of edge devices and enabling real-time responses of critical services, such as autonomous driving (Loghin et al., 2019; Wang et al., 2020). However, literature on hybrid edge-cloud computing for machine learning models is relatively scarce. To our knowledge, the most relevant topic in the literature is split computing, which involves partitioning modules of machine learning pipelines or layers of neural network models between edge and cloud devices to balance overall computation cost and efficiency (Matsubara et al., 2022; Osia et al., 2020). Communication between the edge and the cloud in split computing is inherently synchronized, as both devices contribute to completing one inference run. Another notable paradigm for hybrid computing in machine learning is federated learning, which leverages multiple computing devices for training machine learning models for safety or efficiency purposes (Bonawitz et al., 2019). However, this technique is less commonly used for inference. Cloud service providers such as AWS also have developed patterns for hosting machine learning pipelines across local and cloud devices (AWS-Whitepaper, 2021). The design usually involves splitting the components of a machine learning pipeline, with the core machine learning models still hosted in the cloud. In addition to hybrid computing, there is also literature on improving the efficiency of models deployed on edge devices (Tambe et al., 2021) as well as methods focused on reducing the size of large models for deployment on smaller devices (Hoefler et al., 2021). These methods are orthogonal to our work.

**Retrieval Augmented Models**  Retrieval augmentation is a technique that enhances a language model with retrieved information from external databases. Various methods have been proposed to integrate the retrieved data into the language model, including the use of prompts (Lewis et al., 2020; Guu et al., 2020; Shi et al., 2023), cross-attention modules (Borgeaud et al., 2021), vector concatenation (Izacard & Grave, 2021; Fan et al., 2021), and output distribution adjustment at decoding (Khandelwal et al., 2020; Liu et al., 2022). In our HybridRAG work, we adopt the prompting method, which incorporates retrieved data into the input. However, the HybridRAG framework can be extended to other retrieval augmentation approaches.

---

**Algorithm 1** Inference workflow of HybridRAG

---

**Require:** current user input $\boldsymbol{x}_t$, input history $\boldsymbol{x}_{t-1}$, a retrieval corpus $\mathcal{D}$, a retrieval model $\mathcal{M}_{\text{retrieval}}$, a cloud-based LLM $\mathcal{M}_{\text{cloud}}$, a client language model $\mathcal{M}_{\text{client}}$, a memory $\mathcal{M}$

    **while** $\boldsymbol{x}_t$ **do**
        Compute the changes in context $\text{ED}_t = \text{EditDistance}(\boldsymbol{x}_t, \boldsymbol{x}_{t-1})$
        **if** $\text{ED}_t > \tau$ **then**                 ▷ Send async request to the cloud
            **async** Retrieve relevant documents $\mathcal{D}_r = \{d_1, ...d_k\}$: $\mathcal{D}_r \sim \mathcal{M}_{\text{retrieval}}(\boldsymbol{x}_t, \mathcal{D})$
            **async** Generate memory $m_t \sim \mathcal{M}_{\text{cloud}}(\mathcal{D}_r)$
            Update $\mathcal{M}$ with $m_t$: $\mathcal{M} = Update(\mathcal{M}, m_t)$
        **end if**
        Sample $\boldsymbol{y}_t \sim \mathcal{M}_{\text{client}}(\boldsymbol{x}_t, \mathcal{M})$         ▷ Text prediction with the client side model
        **if** $Accept(\boldsymbol{y}_t)$ **then**                   ▷ User accepts suggestion
            $\boldsymbol{x}_{t-1} \leftarrow \{\boldsymbol{x}_{t-1}, \boldsymbol{x}_t\}, \boldsymbol{x}_t \leftarrow \{\boldsymbol{x}_t, \boldsymbol{y}_t\}$
        **else**
            $\boldsymbol{x}_t \leftarrow \{\boldsymbol{x}_t, Input()\}$      ▷ User rejects suggestion and continues to enter new input
        **end if**
    **end while**

---

# 3   Hybrid Retrieval Augmented Generation

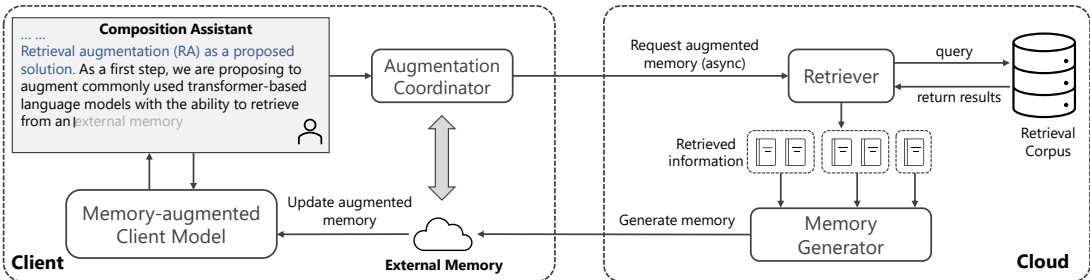

Figure 1: Overview of the HybridRAG framework

We present our HybridRAG approach that leverages cloud-generated memory to enhance the utility of client-based language model while maintaining low latency. The HybridRAG framework consists of four main components: an augmentation coordinator (client), a memory-augmented client model (client), a retriever model (cloud), a memory generator (cloud). Figure 1 illustrates the model architecture. The augmentation coordinator monitors the writing context and determines when to request an augmented memory from the cloud. The retriever model on the cloud server then searches the retrieval corpus to find relevant data. Subsequently, the memory generator employs the GPT-3 model to construct an augmented memory that includes all essential information from the retrieved data, optimizing its usefulness. Finally, the augmented memory is transmitted to the client and seamlessly integrated into the client model, thereby enhancing its overall performance. Algorithm 1 describes the inference workflow of HybridRAG.

In the following subsections, we discuss the details of the different components.

## 3.1   Augmentation Coordinator

The augmentation coordinator component is responsible for managing the augmented memory $\mathcal{M}$ by monitoring changes to the writing context. The function of the augmentation coordinator is depicted in Figure 2. To determine whether a memory update is necessary, the coordinator takes into account both the current context $\boldsymbol{x}_t$ and the context $\boldsymbol{x}_{t-1}$ from the previous step and calculates the context edit distance $\text{ED}(\boldsymbol{x}_t, \boldsymbol{x}_{t-1})$. Once the distance exceeds a pre-determined threshold $\tau$, the coordinator initiates a request to the cloud server for augmented memory. We employ the Levenshtein distance (Yujian & Bo, 2007) to measure the token-level difference. To avoid redundant memory requests, we adopt an incremental memory update approach, where only the newly updated context is used as the query input to generate the new memory $m_t$. When the augmented memory reaches

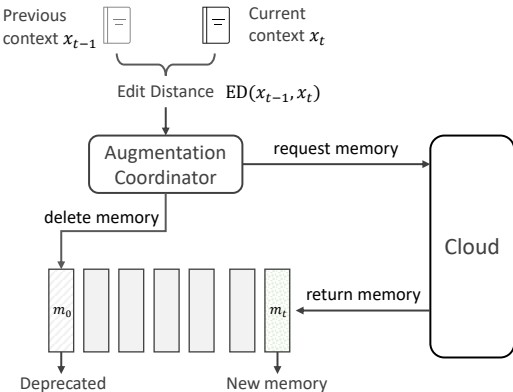

Figure 2: Process of the augmentation coordinator

its maximum capacity of $\mathcal{M}$, the earliest stored memory is replaced with the new one. This process is depicted in Figure 2, where we observe that upon reaching the maximum memory capacity, the oldest memory $m_0$ is swapped out and replaced by the new memory $m_t$.

## 3.2 RETRIEVAL-AUGMENTED MEMORY GENERATOR

Upon receiving a request from the augmentation coordinator, the memory generator on the cloud initiates the preparation of the augmented memory, which will be returned to the client. The memory preparation process consists of two steps: document retrieval and memory generation.

**Document Retrieval**    Given an input query $x$, the goal of the retriever is to select the top-k most relevant documents $\mathcal{D}_r = \{d_1, \ldots, d_k\}$ from a large retrieval corpus $\mathcal{D}$, where $\mathcal{D}_r \subseteq \mathcal{D}$. Following prior work (Lewis et al., 2020; Ni et al., 2021), we use the Dense Passage Retrieval (DPR) (Karpukhin et al., 2020) method, which is a dual encoder retrieval model pre-trained for question and answering task. DPR encodes the document and the query with a context encoder and a query encoder respectively, and calculates the cosine similarity of the encoded embeddings to retrieve the top-k most relevant documents.

**Memory Generation**    After retrieving the relevant documents $\mathcal{D}_r$, instead of directly concatenating them to the original prompt as in Retrieval Augmented Generation (RAG) (Lewis et al., 2020), we employ a LLM to generate concise bullet points that capture the essential information from the retrieved documents. We hypothesize that the improved representation of the memory could enhance the performance of the client model, which is a smaller language model that usually struggles with processing long context. In addition, extracting the key takeaways significantly reduces the memory size, resulting in lower communication and inference cost for the client.

To generate concise bullet points from retrieved documents $\mathcal{D}_r$, we first split the documents into text chunks $\{p_1, \ldots, p_l\}$, where $l$ is the number of chunks. We choose an appropriate chunk size that maintains sentence integrity, avoiding breaking sentences in the middle. Once the chunks are created, we utilize a LLM to extract the key takeaways from each chunk $p_i$. To minimize the frequency of LLM call requests, we consolidate multiple chunks within one document. We show an example of bullet points generated for two text chunks in Appendix A. Subsequently, all the generated bullet points from the retrieval documents are merged to form the memory $m_t$ for the current $t$-th memory request. This memory is then combined with the existing memory to construct the new $\mathcal{M}$ by the augmentation coordinator.

## 3.3 MEMORY-AUGMENTED CLIENT MODEL

While most previous work on memory-augmented networks (Weston et al., 2014; Sukhbaatar et al., 2015; Yogatama et al., 2021; Wu et al., 2022) focus on language modeling task performance, our

client model aims to effectively leverage the augmented memory generated by the LLM. We hypothesize that although a small language model may lack the capacity to handle complex tasks like its larger counterpart, it can still be trained effectively to leverage augmented memory to accomplish simpler tasks, such as paraphrasing content. To this end, we propose an instruction-finetuning approach that aims to bolster the client model's ability to effectively leverage augmented memory.

Training such a client model is a challenging task due to the scarcity of training data. Specifically, it requires data that contains triplets of input prompt $x$, augmented memory $\mathcal{M}$, and reference label for text completion $\hat{y}$. However, obtaining the latter two can be difficult. To address this, we propose a novel method that leverages a LLM to generate the necessary training data.

Given a document $d$, we select a percentage of the document to serve as the input prompt $x = \mathcal{I}(d)$. Then we generate the augmented memory $\mathcal{M}$ with the steps outlined in Section 3.2. As for the reference labels, a straightforward approach is to directly use the remaining part of the document $d$. However, this is no ideal since the original text may not encompass the information contained in the augmented memory. The discrepancy between the completion and the augmented memory can negatively impact the performance of the client model. To address this issue, we employ the LLM to generate the text completion labels. We structure the input prompt and augmented memory into an instruction-based prompt following the format specified in Table 1. This enables us to instruct the LLM to complete the input text based on the provided memory. The completion label can be expressed as $\hat{y} = \mathcal{M}_{\text{cloud}}(\mathcal{I}(d), \mathcal{M})$, with $\mathcal{M}_{\text{cloud}}$ referring to the LLM model for generating the reference label. Additional details can be found in Appendix D, where we present empirical evidence demonstrating that LLM-completed labels outperform ground truth labels from the original text.

After preparing the training data, we proceed to finetune our client model using the instruction-enhanced prompt along with the LLM-generated reference labels. With instruction-tuning, the client model learns to effectively utilize the augmented memory generated by the LLM. To minimize the discrepancy between our model's predictions and the LLM-completed reference labels, we employ the cross-entropy loss on the generated tokens, as defined in Equation 1.

$$\mathcal{L}_d = -\sum_{i=1}^{l} \hat{y}_i \log \left( p_\theta(y_i | \boldsymbol{x}, \hat{y}_{<i}) \right), \tag{1}$$

where $l$ is the length of reference label and $p_\theta(\cdot)$ refers to the probability of tokens generated by the client model.

Table 1: Example of constructing an instruction-enhanced prompt for reference label generation

| Prompt | *Reference*: In 2020, Generative Pre-trained Transformer 3 (GPT-3) was unveiled, a deep learning-based autoregressive language model that can produce human-like text. When provided an initial text as a prompt, it can then generate text that follows on from it. ... This process has eliminated the need for laborious manual labeling and human supervision. 
 *Complete the following text based on the reference:* 
 Generative Pre-trained Transformer 3 (GPT-3) is an autoregressive language model released in 2020 that |
|---|---|
| Output | is capable of producing human-like text when prompted with an initial text. GPT-3 has a 2048-token-long context, a record-breaking 175 billion parameters and a storage capacity of 800GB. |

## 4 EXPERIMENTS

In this section, we present the evaluation results of our proposed HybridRAG approach on multiple benchmark datasets. We introduce the experiment setup in Section 4.1 and detail the performance of the proposed method compared against several baselines in terms of composition assistance utility and inference latency in Section 4.2. Furthermore, we present a case study and describe limitations of our approach in Section 4.3. Additionally, we provide further results in the appendix, including ablation studies on retrieval sources and number of retrieved documents.

### 4.1 EXPERIMENTAL SETUP

**Datasets and Labels** We evaluate our framework on the WikiText-103 dataset (Merity et al., 2016) and four datasets from the Pile benchmark (Gao et al., 2020). For instruction-tuning the client model,

we use the training set of WikiText-103, which consists of 15,220 Wikipedia passages. We process the first section of each passage for text completion by randomly selecting a prompt length between [16, 64] tokens and using the first few tokens of that length as the input prompt. We then retrieve from the remaining sections of the passages for memory generation. We also provide more results of retrieving from the entire WikiText dataset in the appendix. For evaluation, we use the WikiText-103 test set and four subsets from the Pile benchmark: Enron Emails, HackerNews, NIH ExPorter, and Youtube Subtitles. These datasets encompass a diverse range of domains, including news, emails, medical documents and video subtitles. On Pile datasets, we process the first paragraph of the data for text prediction and use the remaining paragraphs for retrieval. We use a LLM to generate reference label and we evaluate our model completion against LLM generated completions.

**Evaluation Metrics**  We employ several automated metrics to evaluate the model's utility. We calculate the perplexity (Jelinek et al., 1977) of the model by measuring how well it predicts the reference labels based on the prompts.

$$\text{PPL} = \exp\left(-\frac{1}{l}\sum_{i=1}^{l} log(p_\theta(\hat{\boldsymbol{y}}|\boldsymbol{x}))\right) \tag{2}$$

Perplexity indicates the language model's level of uncertainty when processing the given text, with lower perplexity indicating higher model confidence of observing the reference label. In addition, we also use lexical and semantic similarity metrics, GLEU (Wu et al., 2016), BLEU (Papineni et al., 2002), ROUGE (Lin, 2004), METEOR (Banerjee & Lavie, 2005), and BERTScore (Zhang et al., 2020), to evaluate the degree of similarity between the model's predictions and the reference.

To evaluate the inference latency of our system, we measure the average running time required for three steps in our framework: document retrieval, memory generation, and text prediction. This allows us to quantify the time cost associated with each of these steps and analyze the overall efficiency of our system.

**Implementation Details**  To implement our client model, we compare two OPT language models (Zhang et al., 2022): OPT-125M and OPT-350M. Both models are decoder-only pre-trained transformers that have 125 million and 350 million parameters respectively, which are small enough to be well-suited for real-time composition assistance. We employ greedy search for client model decoding. For the cloud-based LLM, we use the GPT-3 Davinci model from the OpenAI API[2], and set $\text{temperature} = 0$, $\text{top\_}p = 1$ to make the generation more deterministic. We set the maximum output tokens to 64 for both reference label generation and text prediction. For document retrieval, we process the retrieval text into chunks of 128 tokens and use the DPR model and the Faiss library (Johnson et al., 2019) for efficient retrieval.

The client models are trained on machines equipped with one Tesla V100 GPU with 16GB memory. For latency evaluation, we deploy the client models on two different machines: a GPU machine with an 11GB Nvidia Tesla K80 GPU, and a laptop without a GPU (specifically a Surface 4 laptop featuring an Intel i7 CPU @3.00GHz with 4 cores and 16GB of physical memory). We set the maximum output tokens to 15 for latency evaluation.

**Baseline Methods**  We compare our approach against the following baselines:

1. **Vanilla OPT**: We employ a vanilla client OPT model for text completion, which does not use any additional memory or assistance from the cloud.
2. **RAG**: The RAG approach can be easily turned into a hybrid model with our framework. In this setting, we use the DPR model to retrieve relevant text from the cloud and feed the full retrieved text to the client model for generation.
3. **HybridRAG without finetuning (HybridRAG w/o FT)**: To assess the efficacy of our instruction-tuned client model, we examine a HybridRAG model without applying finetuning to the client model for text prediction.
4. **GPT-3 zero-shot**: We use the GPT-3 Davinci model in a zero-shot manner for text completion. However, it's important to note that the GPT-3 model cannot be deployed on client devices for real-time composition assistance.

---

[2]https://platform.openai.com/docs/models/gpt-3

Table 2: Peformance comparison of HybridRAG models and baselines on the Wikitext-103 dataset

|  |  | PPL | GLEU | BLEU-4 | ROUGE-1 | ROUGE-L | METEOR | BERTScore |
|---|---|---|---|---|---|---|---|---|
|  | GPT-3 zero-shot | 4.9 | 32.4 | 31.3 | 50.0 | 44.3 | 43.6 | 89.4 |
| OPT-125M | Vanilla OPT | 10.3 | 12.5 | 8.6 | 28.9 | 23.7 | 22.0 | 84.2 |
|  | RAG | 6.7 | 15.5 | 12.6 | 32.4 | 27.0 | 25.9 | 84.9 |
|  | HybridRAG w/o FT | 3.3 | 31.1 | 31.1 | 45.8 | 40.7 | 40.5 | 87.7 |
|  | HybridRAG | **3.3** | **34.1** | **33.9** | **48.4** | **42.9** | **43.4** | **88.4** |
| OPT-350M | Vanilla OPT | 8.1 | 14.0 | 10.2 | 31.3 | 25.5 | 24.3 | 84.9 |
|  | RAG | 5.3 | 18.3 | 15.8 | 35.5 | 29.4 | 29.1 | 85.8 |
|  | HybridRAG w/o FT | 2.9 | 34.6 | 35.0 | 49.0 | 43.8 | 44.0 | 88.5 |
|  | HybridRAG | **2.7** | **37.8** | **37.8** | **52.4** | **47.2** | **47.5** | **89.3** |

When evaluating the baseline models, we ensure a fair comparison by regenerating reference labels for each baseline using the GPT-3 model, based on the memory used by that baseline. Specifically, for the Vanilla OPT baseline, reference labels are generated with GPT-3 without additional memory. For RAG, reference labels are generated by GPT-3 with full text. In the case of GPT-3-zeroshot baseline, since there is no ideal reference label for comparison, we used the same label as our HybridRAG approach.

## 4.2 EXPERIMENTAL RESULTS

### 4.2.1 UTILITY

Table 2 presents the utility of our models compared to the baselines on the Wikitext-103 dataset, with respective results for OPT-125M and OPT-350M models. The results demonstrate that our approach outperforms all client-only and hybrid baselines across all evaluated metrics on Wikitext-103. Compared to the vanilla OPT models, the HybridRAG models exhibited remarkable performance improvements. On average, the HybridRAG approach achieves an improvement of 67.6% in perplexity and 171.7% in GLEU across OPT-125M and OPT-350M. By comparing the RAG approach, where the retrieved documents are fed to the client model in its original text form, and HybridRAG w/o FT, we can observe that the utilization of LLM-compressed memory leads to a significant average performance gain of 48.6% in GLEU. It shows that the representation of the memory is vital to the client model performance. Furthermore, when comparing our full approach to the variant without finetuning, our model outperformed it by 4.7% and 9.5% in the respective metrics, which indicates that instruction-tuning helps the model to better leverage the context. The GPT-3 model demonstrated impressive zero-shot performance without any additional context. Given that both GPT-3 and OPT models are likely trained on Wikipedia, it's probable that the larger GPT-3 model utilized its parametric memory for generation more than the smaller OPT models. Nevertheless, the results indicate that HybridRAG can greatly improve the performance of the client model, to the extent that OPT with HybridRAG can rival GPT-3's zero-shot performance with the help of relevant hybrid augmented memory.

Table 3 presents the perplexity and GLEU results for the four Pile subsets, with additional metrics in Appendix E. Consistent with the findings on the Wikitext-103 dataset, our model demonstrates better performance compared to the baseline models across all four datasets. It is important to note that we did not finetune the client model specifically on the Pile datasets, which suggests the model's generalization capabilities. We have also observed a high perplexity for the zero-shot GPT-3 baseline. This is due to that the reference label used for this baseline is generated with the augmented memory, which is not seen by the GPT-3 zero-shot baseline, and therefore a distribution shift.

### 4.2.2 INFERENCE LATENCY

We performed a latency evaluation for both the OPT-125M and OPT-350M models on the two hardware setups, as described in Section 4.1. Figure 3a shows the run times for the client model on the GPU machine. It indicates that the OPT-125M model exhibits a 49.3% faster inference time compared to the OPT-350M model. This finding emphasizes that the size of the client model plays a crucial role in the inference time. Figure 3b presents the run time for the retrieval and memory gener-

Table 3: Performance comparison of HybridRAG models and baselines on the Pile subsets

|  |  | Enron Emails | | NIH ExPorter | | Hacker News | | Youtube Subtitles | |
|---|---|---|---|---|---|---|---|---|---|
|  |  | PPL | GLEU | PPL | GLEU | PPL | GLEU | PPL | GLEU |
|  | GPT-3 zero-shot | 106.9 | 12.3 | 12.2 | 18.5 | 65.1 | 15.3 | 36.6 | 13.7 |
| OPT-125M | Vanilla OPT | 6.0 | 10.5 | 5.4 | 12.0 | 7.8 | 11.8 | 6.4 | 9.4 |
|  | RAG | 3.7 | 12.7 | 3.8 | 11.5 | 4.7 | 15.8 | 4.5 | 12.2 |
|  | HybridRAG w/o FT | **3.2** | **20.3** | **2.9** | 19.5 | **3.8** | 19.5 | **3.1** | 15.9 |
|  | HybridRAG | 3.7 | 18.9 | 3.3 | **23.0** | 4.0 | **21.7** | 3.5 | **17.1** |
| OPT-350M | Vanilla OPT | 5.4 | 8.7 | 6.3 | 13.1 | 4.8 | 12.7 | 5.5 | 9.7 |
|  | RAG | 3.4 | 13.6 | 3.9 | 17.9 | 3.3 | 13.0 | 3.8 | 14.8 |
|  | HybridRAG w/o FT | **2.9** | 19.9 | 3.3 | 22.1 | **2.5** | 23.6 | **2.8** | 20.1 |
|  | HybridRAG | **2.9** | **23.6** | **3.0** | **24.0** | 2.7 | **25.7** | **2.8** | **21.7** |

ation steps. The results indicate that memory generation utilizing a large language model consumes the majority of the memory preparation time. Figure 3c compares our asynchronous HybridRAG approach with OPT-125M to a synchronous inference approach by directly calling a GPT-3 model and a retriever for composition assistance. Notably, our approach showcases an impressive speed enhancement, achieving a remarkable 138.3 times faster performance compared to the synchronous approach. Lastly, we conducted a comparison of the running time of HybridRAG OPT-125M between the GPU machine and the laptop in Figure 3d. The results indicate that our approach can be deployed on user edge devices without GPUs, although the inference time is approximately 1.45 times slower compared to a GPU machine. It should be noted that we didn't optimize the client model for decoding speed with caching or quantization. These methods are orthogonal to our work and can be used in conjunction with our approach to further reduce the inference latency.

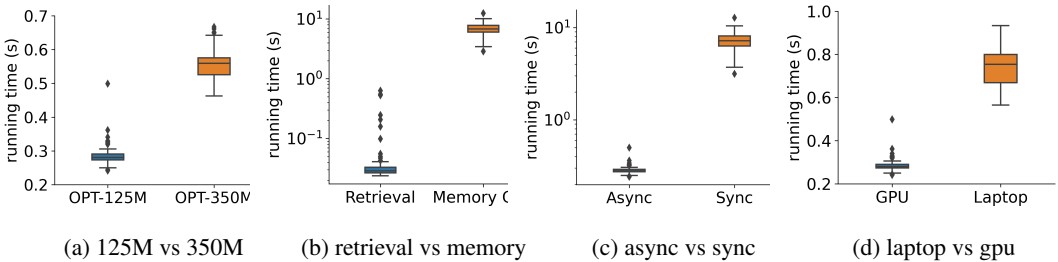

| (a) 125M vs 350M | (b) retrieval vs memory | (c) async vs sync | (d) laptop vs gpu |
|---|---|---|---|

Figure 3: Inference latency for client inference, retrieval and memory generation on multiple devices

### 4.2.3 ASYNCHRONOUS MEMORY UPDATE

Figure 5 illustrates the impact of asynchronous memory update on model utility. To measure this effect, we conducted an experiment in which we gradually increased the edit distance threshold that determines how often the client model requests for memory updates. Figure 4 shows an example of how we set the edit distance threshold. For a given prompt, we use the initial part of the prompt as the query for memory generation and use the full prompt for text prediction.

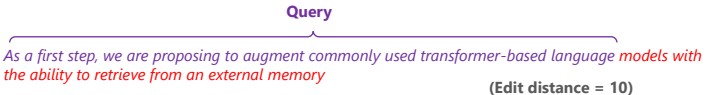

Figure 4: An example of setting edit distance threshold = 10.

As expected, as the edit distance threshold increases, the memory becomes less up-to-date due to the increased difference between the query used for memory generation and current input context for text completion, resulting in a decline in model utility. As the threshold reaches 20, we observe a notable drop in the GLEU score compared to synchronous memory update for both OPT-125M and OPT-350M. Nevertheless, it still significantly outperformed the vanilla OPT baselines.

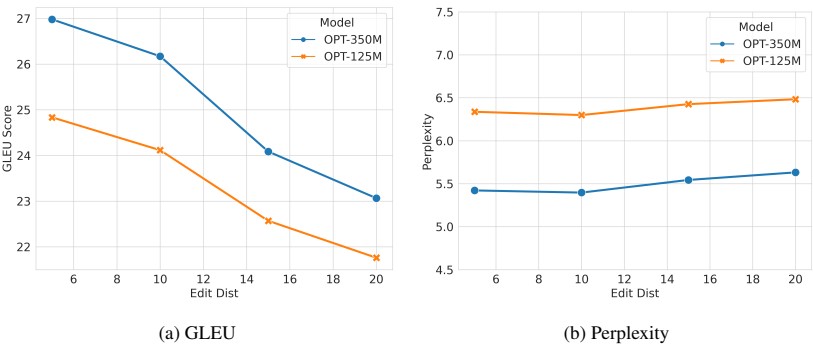

|     |     |
| --- | --- |
| (a) GLEU | (b) Perplexity |

Figure 5: HybridRAG performance with asynchronous memory update

Table 4: Completions of the prompt by different models, truncated to the first sentence.

| prompt | Graham Arthur Chapman (8 January 1941 - 4 October 1989 |
| --- | --- |
| GPT-3 zero-shot | ) was an English cricketer. |
| GPT-3-generated label | ) was an English comedian, writer, actor, author, and one of the six members of the British surreal comedy group Monty Python. |
| HybridRAG OPT-125M | ) was a British actor and comedian. |
| Vanilla OPT-350M | ) was an English cricketer. |
| HybridRAG OPT-350M | ) was a British comedian and actor. |

## 4.3 CASE STUDY AND LIMITATIONS

To better understand the strengths and limitations of HybridRAG, we manually examined the completions of different models. Table 4 shows an example of HybridRAG generating better completions, thanks to its ability to leverage cloud-based resources. However, we also identified cases where the model failed to produce good completions, which are shown in Appendix F. We find that the performance of the client model highly depends on the memory. We have also observed cases where the client model combines information from different bits of the memories, resulting in fabrication of inaccurate information. In addition, the larger GPT-3 model can ignore the memory and use its parametric knowledge for generation when the augmented memory deviates from the input prompt, whereas the smaller client model tends to adhere to the memory content. Improving the memory generator by reducing duplicate information, and enhancing the reasoning abilities of the client model or encouraging it to stick to the memories content would be some of the directions to address these failing cases and limitations.

## 5 CONCLUSION

In this paper, we propose HybridRAG, a novel hybrid retrieval-augmented generation approach for real-time composition assistance. By integrating LLM-enhanced memory into our instruction-tuned client model with asynchronous update, we show with experiment results on multiple datasets that our hybrid retrieval approach enables substantial utility improvements over smaller language models while maintaining inference efficiency, making it a valuable solution for real-time tasks.

In our work, we employ retrieval-based memory augmentation as the solution to combine the powerful LLM on the cloud and the more agile client model. Naturally, the performance of the system relies on the quality of the memory that the cloud provides to the client and how the memory is integrated into the client model. The quality of the memory is influenced by multiple factors: the representation of the memory (e.g. original text vs condensed information snippets), the relevance of the retrieved data, and the freshness of information compared to the current input context. In future work, we will continue to investigate more effective ways of refining memory according to the tasks and explore alternative memory augmentation approaches to further enhance our model performance.

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

## A   EXAMPLE OF MEMORY GENERATION

Table 5 shows an example of augmented memory generated by GPT-3.

Table 5: Example of Augmented Memory Generated by GPT-3 Model.

| Prompt | Generate a list of key takeaways in complete sentences for each paragraph one by one, containing all the details such as entities, persons, organizations, numbers, years, and locations. 
 P1: *Generative Pre-trained Transformer 3 (GPT-3) is an autoregressive language model released in 2020 that uses deep learning to produce human-like text. Given an initial text as prompt, it will produce text that continues the prompt.* 
 P2: *The architecture is a decoder-only transformer network with a 2048-token-long context and then-unprecedented size of 175 billion parameters, requiring 800GB to store. ... This eliminated the need for human supervision and for time-intensive hand-labeling.* 
 Key Takeaways: ### P1: |
|---|---|
| Output | *- Generative Pre-trained Transformer 3 (GPT-3) is a deep learning-based autoregressive language model released in 2020 that produces human-like text.* 
 *- It takes an initial text as prompt and produces text that continues the prompt.* 
 ### P2: 
 *- GPT-3 has a decoder-only transformer network with a 2048-token-long context and 175 billion parameters, requiring 800GB to store.* 
 *- It is trained to predict what the next token is based on previous tokens using generative pre-training. ...* |

## B   LOCAL VS. GLOBAL RETRIEVAL CORPUS

Figure 6 displays a performance comparison between local and global retrieval corpus on the Wikitext-103 dataset. In the local setting, the retrieval corpus consists of the texts contained within the same document, while in the global setting, the entire dataset is utilized as the retrieval corpus. The results indicate that the local retrieval corpus yields superior GLEU and Perplexity scores. This can be attributed to the smaller pool of documents in the local corpus, making it easier to obtain the relevant documents for retrieval.

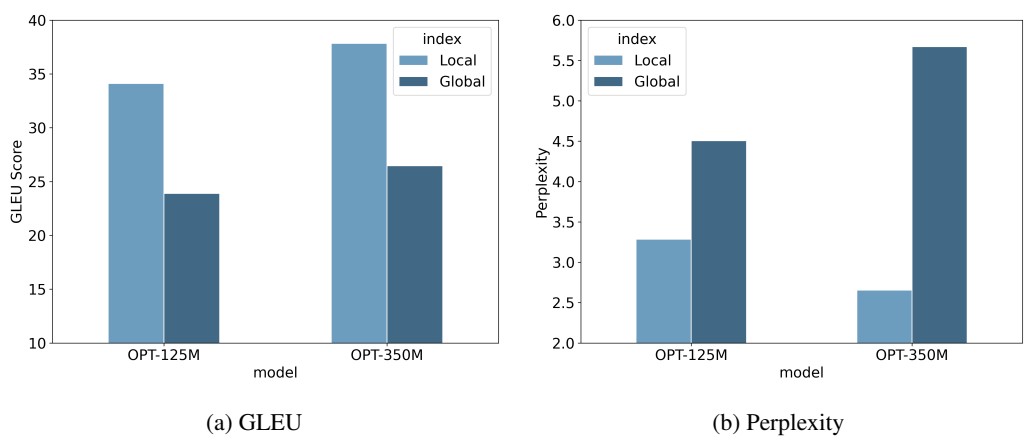

(a) GLEU

(b) Perplexity

Figure 6: Comparison between local and global retrieval corpus

## C   NUMBER OF RETRIEVAL DOCUMENTS

Figure 7 depicts the model performance for various numbers of retrieval documents on Wikitext-103 dataset. Based on the results, we can deduce that both the OPT-125M and OPT-350M models exhibit optimal performance when four retrieval documents are used. As more documents are included beyond four, the performance remains consistent as the top four documents already encompass the majority of relevant information for our task.

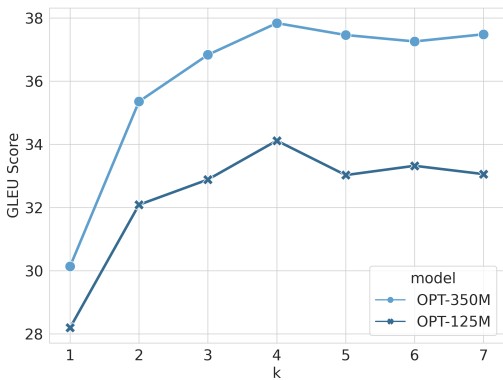

Figure 7: Performance analysis on number of retrieval documents

## D    LLM COMPLETION LABELS

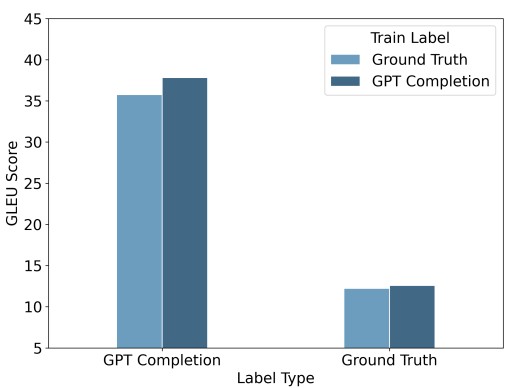

Figure 8: Comparison between ground truth labels derived from GPT-3 completion and the original text

We investigate the discrepancy between using ground-truth labels from GPT-3 completion and the original text for both model training and evaluation. As shown in Figure 8, fine-tuning our client model on GPT3 completion labels consistently yields consistent better results. Notably, even when evaluating with the original text as ground truth, the model fine-tuned with GPT3-generated labels outperforms the alternative.

## E    MORE RESULTS ON PILE DATASETS

The results of the four Pile datasets on all seven metrics are presented in Tables 6 and 7. We can observe that our model consistently outperforms all the other baselines, demonstrating the superior performance. However, it is worth noting that the perplexity of the OPT-125M model does not consistently surpass that of the HybridRAG w/o FT model.

## F    MORE EXAMPLES FOR CASE STUDY

Table 8 shows another working example for HybridRAG models and Table 9 shows an example of a failing case for both OPT-125M and OPT-350M HybridRAG models. The OPT-125M model generates a repetitive and factually incorrect text, while the OPT-350M model generates a text that is factually incorrect. GPT-3 Davinci, however, can still use the same retrieved memories to provide

| | | PPL | GLEU | BLEU-4 | ROUGE-1 | ROUGE-L | METEOR | BERTScore |
|---|---|---|---|---|---|---|---|---|
| Enron Emails | GPT3 zero-shot | 106.9 | 12.3 | 10.4 | 26.1 | 23.3 | 21.6 | 83.6 |
| | Vanilla OPT | 5.4 | 8.7 | 6.5 | 21.1 | 18.6 | 17.1 | 80.2 |
| | RAG | 3.4 | 13.6 | 12.2 | 26.8 | 24.0 | 22.9 | 81.2 |
| | HybridRAG w/o FT | **2.9** | 19.9 | 19.4 | 32.7 | 30.2 | 28.8 | 83.0 |
| | HybridRAG | **2.9** | **23.6** | **23.7** | **35.0** | **32.3** | **31.9** | **84.7** |
| NIH ExPorter | GPT3 zero-shot | 12.2 | 18.5 | 16.2 | 36.6 | 31.7 | 29.2 | 86.7 |
| | Vanilla OPT | 6.3 | 13.1 | 8.9 | 31.9 | 26.7 | 23.9 | 85.3 |
| | RAG | 3.9 | 17.9 | 15.2 | 36.7 | 31.1 | 29.7 | 86.5 |
| | HybridRAG w/o FT | 3.3 | 22.1 | 20.2 | 40.0 | 35.1 | 33.5 | **87.1** |
| | HybridRAG | **3.0** | **24.0** | **22.9** | **41.5** | **37.1** | **35.4** | 87.1 |
| Hacker News | GPT3 zero-shot | 65.1 | 15.3 | 14.3 | 30.2 | 27.7 | 20.4 | 85.8 |
| | Vanilla OPT | 4.8 | 12.7 | 11.3 | 29.3 | 26.4 | 22.4 | 84.9 |
| | RAG | 3.3 | 13.0 | 12.2 | 26.8 | 24.5 | 20.5 | 84.3 |
| | HybridRAG w/o FT | **2.5** | 23.6 | 24.4 | 37.9 | 35.7 | 31.5 | **86.6** |
| | HybridRAG | 2.7 | **25.7** | **25.9** | **39.5** | **36.8** | **34.4** | 86.5 |
| Youtube Subtitles | GPT3 zero-shot | 36.6 | 13.7 | 11.8 | 27.1 | 24.5 | 23.3 | 84.4 |
| | Vanilla OPT | 5.5 | 9.7 | 7.2 | 22.2 | 20.0 | 19.1 | 82.4 |
| | RAG | 3.8 | 14.8 | 12.9 | 28.4 | 25.3 | 25.1 | 84.7 |
| | HybridRAG w/o FT | 2.8 | 20.1 | 19.9 | 32.9 | 30.4 | 29.7 | 85.1 |
| | HybridRAG | **2.8** | **21.7** | **21.1** | **34.6** | **31.8** | **31.9** | **85.9** |

Table 6: Comparison of the utility performance of the OPT-350M-based HybridRAG models and baselines on Pile datasets

| | | PPL | GLEU | BLEU-4 | ROUGE-1 | ROUGE-L | METEOR | BERTScore |
|---|---|---|---|---|---|---|---|---|
| Enron Emails | GPT3 zero-shot | 106.9 | 12.3 | 10.4 | 26.1 | 23.3 | 21.6 | 83.6 |
| | Vanilla OPT | 6.0 | 10.5 | 9.1 | 21.5 | 19.3 | 18.0 | 80.3 |
| | RAG | 3.7 | 12.7 | 11.9 | 25.2 | 22.9 | 21.6 | 80.4 |
| | HybridRAG w/o FT | **3.2** | **20.3** | **19.9** | 31.0 | **28.6** | 27.4 | 82.7 |
| | HybridRAG | 3.7 | 18.9 | 18.9 | **31.6** | 28.3 | **28.2** | **83.8** |
| NIH ExPorter | GPT3 zero-shot | 12.2 | 18.5 | 16.2 | 36.6 | 31.7 | 29.2 | 86.7 |
| | Vanilla OPT | 5.4 | 12.0 | 10.8 | 27.9 | 25.3 | 21.4 | 84.3 |
| | RAG | 3.8 | 11.5 | 10.5 | 25.4 | 23.3 | 18.6 | 83.8 |
| | HybridRAG w/o FT | **2.9** | 19.5 | 19.8 | 33.5 | 31.4 | 27.2 | 85.6 |
| | HybridRAG | 3.3 | **23.0** | **23.5** | **36.2** | **33.2** | **30.8** | **85.9** |
| Hacker News | GPT3 zero-shot | 65.1 | 15.3 | 14.3 | 30.2 | 27.7 | 20.4 | 85.8 |
| | Vanilla OPT | 7.8 | 11.8 | 7.5 | 29.7 | 25.2 | 22.2 | 84.7 |
| | RAG | 4.7 | 15.8 | 12.8 | 34.1 | 29.1 | 27.2 | 85.8 |
| | HybridRAG w/o FT | **3.8** | 19.5 | 17.5 | 37.3 | 32.6 | 30.8 | 86.4 |
| | HybridRAG | 4.0 | **21.7** | **20.5** | **38.0** | **32.8** | **32.2** | **86.5** |
| Youtube Subtitles | GPT3 zero-shot | 36.6 | 13.7 | 11.8 | 27.1 | 24.5 | 23.3 | 84.4 |
| | Vanilla OPT | 6.4 | 9.4 | 6.9 | 23.2 | 20.7 | 19.6 | 82.9 |
| | RAG | 4.5 | 12.2 | 9.6 | 26.1 | 23.2 | 23.4 | 84.1 |
| | HybridRAG w/o FT | **3.1** | 15.9 | 15.0 | 28.5 | 26.1 | 25.1 | 84.2 |
| | HybridRAG | 3.5 | **17.1** | **16.2** | **29.4** | **26.5** | **25.7** | **84.7** |

Table 7: Comparison of the utility performance of the OPT-125M-based HybridRAG models and baselines on Pile datasets

a factual and useful completion for this prompt. The memories are bullet points generated from several document chunks; OPT-125M and OPT-350M limited reasoning abilities does not allow them to process the memories extensively and reorganize them. The working cases correspond to situations where the client model only has to paste an existing chunk of text as is; we've noticed that when the small OPT client models combine information from different bits of the memories, they usually end up fabricating wrong facts. On top of this, the completions from the current finetuned OPT-350M model seem to put more emphasis on the (limited) parametric memory than on the retrieved memories, leading to more "creative" – but factually wrong – generated text.

| prompt | Gears of War 2 is a 2008 military science fiction third-person shooter video |
|---|---|
| GPT3 zero-shot | game developed by Epic Games and published by Microsoft Game Studios for the Xbox 360. It is the second installment of the Gears of War series and the sequel to 2006's Gears of War. The game was released worldwide in |
| GPT3-generated label | game. The title and spoken words of the trailer are taken from the World War I poem Ï Have a Rendezvous with Deathb̈y Alan Seeger. In Gears of War 2, players are able to carry a |
| HybridRAG OPT-125M | game. The title and spoken words of the trailer were taken from Alan Seeger's World War I poem "I Have a Rendezvous with Death". On October 10, 2008, Microsoft and Epic Games announced the game had |
| Vanilla OPT-350M | game developed by Electronic Arts and published by Electronic Arts for the PlayStation 2 and Xbox 360. The game was released on November 26, 2008 for the PlayStation 2 and Xbox 360.
Gameplay
Gears of War |
| HybridRAG OPT-350M | game developed and published by Epic Games and Microsoft. The game is set 18 months after the events of Gears of War 2. The game is set in the fictional world of the United States of America. The player controls a |

Table 8: Completions of the prompt by different models, showcasing a working case for OPT-125M HybridRAG; this is also a working case for OPT-350M HybridRAG if we only consider the first 2 sentences.

| prompt | The Bengali language movement (Bengali: Bhasha Andolon), was a political movement in former East Bengal (today Bangladesh) advocating the recognition of the Bengali language as an |
|---|---|
| GPT3 zero-shot | official language of Pakistan. The movement was a major part of the Bengali Language Movement and led to the establishment of Bengali as one of the state languages of then East Pakistan in 1956.
The movement was |
| GPT3-generated label | official language of the then-Dominion of Pakistan. That sparked many similar movements in other parts of the country, especially in East Pakistan, where many people felt subjugated by the central government dominated by West Pakistani elite. |
| HybridRAG OPT-125M | official language. The movement was founded by the Bengali community in the early 20th century. The movement was founded by the Bengali community in the early 20th century. The movement was founded by the Bengali community |
| Vanilla OPT-350M | official language of the state.
The movement was founded by the Bengali-language activist and writer, Bhasha Andolôn. The movement was founded in the year 2000. The movement was led by the |
| HybridRAG OPT-350M | official language of Bangladesh. The movement was founded in the early 1990s by Fazlur Rahman, a Bengali-language journalist and activist. The movement was founded in the early 1990s by Fazlur |

Table 9: Completions of the prompt by different models corresponding to failing cases for OPT-125M HybridRAG and OPT-350M HybridRAG.

Improving the memory generator by reducing duplicate information, and enhancing the reasoning abilities of the client model or encouraging it to stick to the memories content would be some of the ways to address these failing cases and limitations.

