# OpenReview forum: "Hybrid Retrieval-Augmented Generation for Real-time Composition Assistance"
_ICLR.cc/2024/Conference — ICLR 2024 Conference Withdrawn Submission_

### Official Review · Reviewer_Z6V6 · 2023-11-01

**Soundness:** 3 good
**Presentation:** 2 fair
**Contribution:** 2 fair
**Rating:** 5
**Confidence:** 3

**Summary:**

The authors propose a new framework for a low-latency client-side implementation of a RAG solution that is aided by an asynchronous memory update process that is aided by a cloud LLM, called HybridRAG. The asynchronous nature of the approach allows for a very remarkable speed up in latency. The main (and only?) target task for the approach is composition assistance, which if I understood correctly is text completion aided by a reference corpus.

Their approach consist of 1) a delayed context retriever (the augmentation coordinator), which reduces the number of requests by waiting until a certain "edit distance" threshold is reached. 2) a FIFO memory mechanism; 3) a document retriever using the Dense Passage Retrieval method; 4) a memory generator, which is a cloud-based LLM that summarizes the retrieved documents. They also propose the method for fine-tuning the client LM to be better suited to the HybridRAG task, which I believe is their recommended approach.

Interestingly, they not only improve on the latency of the vanilla RAG approach, they also outperform it in output quality (with both the trained and untrained models).

They also show that even though the delayed update mechanism hurts performance, it is still an improvement over the vanilla baseline.

**Strengths:**

- The method not only improves latency as it originally sets out to, but also seems to improve accuracy. This is fascinating (though I have my reservations about this fact)
- The system design seems simple, yet quite elegant (assuming all the claims are true)
- Even HybridRag "w/o FT" beats the baseline RAG model. This shows great promise (though it also warrants more investigation)

**Weaknesses:**

I have some general reservations about the claims, the most important of which concerns the validity of the dataset and metrics used:
1. the first and main dataset used for evaluation is the WikiText-103 dataset. As the authors also mention, it's highly likely that both the OPT and GPT-3 models have been trained (directly or indirectly) on this or similar data. As such, this makes it an odd choice for a "RAG" method, where the main value prop is improving the model when the base knowledge is lacking. To further support this claim, we also see that "GPT-3 zero-shot" is highly competitive and beats all the models in many of the metrics.

2. GPT-3 is in some sense being used both as the evaluator, the trainer, and the memory generator. The concern with this is the possible bias towards its own results. I believe all the metrics used, e.g. BLEU, ROUGE, BERTScore, etc, are highly susceptible to such bias as they cannot fairly compare two correct solutions that are worded differently.

Here are a set of other more minor issues:
3. HybridRAG w/o finetuning seems pretty much just as good as the finetuned version. So I'm surprised that the authors are actually recommending the finetuned version at all. Also, the fact that this model does so much better than vanilla RAG generates a lot of questions. I would have expected more ablations to get to the bottom of this disparity. Are the summaries really that much more effective? Or is it the FIFO memory mechanism? Why? Looking at the summary samples, they do not look that much more concise. Do you have comparisons on the average length of the documents vs the summarized version?

4. In comparisons (e.g. Table 2), it's not obvious how much "edit distance" or how asynchronous actually are the HybridRAG models. In the absence of this info, makes me believe that they are being compared in the "idealized" scenario, which then makes all latency claims highly unfair.

5. For a method that claims reduced latency as their main benefit, I find the latency comparisons to be not enough, and the existing results seem to be hastily put together.

6. Though I find this point to be slightly unfair, but using the OPT and GPT-3 models seems slightly outdated by now and makes the results less relevant.

7. Lastly, I want to mention that since most of the improvements are on the systems side, I'm not sure if ICLR is the correct venue for this work. I am by no means giving a diminished value to the contributions in this point

**Questions:**

- I'm quite interested in figuring out whether HybridRAG w/o finetune truly did beat vanilla RAG or not. And if your answer is yes, then why can you elaborate why?
- Section 4.2.2 mentions ".. inference time is approximately 1.45 times slower compared to a GPU machine". But eye-balling the numbers from Figure 3.d gives me a larger gap (at least 3x). Am I wrong?

---

### Official Review · Reviewer_wQxv · 2023-11-02

**Soundness:** 2 fair
**Presentation:** 2 fair
**Contribution:** 2 fair
**Rating:** 3
**Confidence:** 4

**Summary:**

The paper presents the Hybrid Retrieval-Augmented Generation (HybridRAG) framework, aimed at enhancing real-time composition assistance. This framework adeptly merges a large cloud-based language model with a client-side model using retrieval-augmented memory. By capitalizing on cloud-generated memory augmentation, HybridRAG amplifies the performance of smaller language models on edge devices, all while functioning asynchronously. Experimental results indicate that HybridRAG markedly surpasses client-only models in performance while maintaining reduced latency.

**Strengths:**

1. The proposed framework facilitates real-time text generation on client devices, harnessing the power of cloud-based retrieval augmentation.

**Weaknesses:**

1. Limited Novelty: The system's enhancement relies on using an LLM to distill essential information from retrieved documents into concise bullet points. Hence, it's unsurprising that HybridRAG performs better given that (1) the LLM effectively stores content into memory and (2) knowledge from the LLM is distilled into smaller models.

2. Restricted Testing Scope: The evaluation is confined to only two datasets - WikiText and Pile. The absence of tests on knowledge-intensive NLP tasks, such as open-domain QA and fact verification, limits its broader applicability. Moreover, the exclusive testing on the OPT model fails to demonstrate the framework's generalizability across other models like LLaMA.

**Questions:**

see weakness

---

### Official Review · Reviewer_VfJr · 2023-11-08

**Soundness:** 3 good
**Presentation:** 3 good
**Contribution:** 3 good
**Rating:** 6
**Confidence:** 4

**Summary:**

In this paper, the authors propose the Hybrid Retrieval-Augmented Generation (HybridRAG) framework, which aims to efficiently combine a cloud-based LLM with a smaller, client-side, language model through retrieval-augmented memory. Specifically, the proposed framework leverages (1) the hybrid retrieval augmentation to enable real-time generation, (2) the LLM-augmented memory approach to improve the utility of the client model, and (3) the augmentation coordinator module to enable asynchronous memory augmentation to minimize the client-to-cloud communication. Experiments on the Wikitext dataset and Pile subsets suggest that HybridRAG improves utility over client-only models while maintaining low latency.

**Strengths:**

- Retrieval augmented generation is an effective method to improve the performance of LLMs but suffers from inefficiency due to the additional retrieval process. Thus the goal of this paper to improve the efficiency of retrieval augmented generation is well-motivated.
- The proposed method is intuitive and easy to understand.

**Weaknesses:**

- The model and baseline used in the paper are slightly outdated. It would be meaningful to see how the proposed method behaves with more recent LLMs and other RAG methods (e.g., LLaMA[1] and RETRO [2], respectively)
- In the paper, the authors claim that the proposed framework can be used for real-time generation. However, the definition of real-time is not clear here and as the context retrieved from the cloud becomes longer, how will this impact the claim on the real-time generation still lacks of evaluation.
- The presentation of the paper can be improved, for example in Fig. 3, there is some overlapping between subfigures, blocking some characters in the label.

[1] Touvron, Hugo, et al. "Llama: Open and efficient foundation language models." arXiv preprint arXiv:2302.13971 (2023).
[2] Borgeaud, Sebastian, et al. "Improving language models by retrieving from trillions of tokens." International conference on machine learning. PMLR, 2022.

**Questions:**

- In Table 3, HybirdRAG w/o FT leads to improved performance over the HybridRAG in some tasks. Could the authors explain why this happens?

---

### Official Review · Reviewer_nsQe · 2023-11-09

**Soundness:** 3 good
**Presentation:** 2 fair
**Contribution:** 3 good
**Rating:** 6
**Confidence:** 3

**Summary:**

This paper presents a framework for text completion on an client device by combining a small edge language model with a cloud-based RAG LLM, showing the framework outperforms simple edge only (and other) approaches.

**Strengths:**

The paper describes and focuses on a interesting problem: text completion in the context of edge devices is very practical and intuitive.

The approach is straightforward and novel. Combining different scales of LLMs for any application is timely with the growing popularity of both open and enterprise LLMs. I particularly find the use of the cloud model to enhance the training of the small model to be particularly compelling.

I like the use of summarized RAG techniques to improve text generation.

The experiments are reasonable and show the approach is promising. The ablated models as baselines clearly show the complexity of the approach is necessary for improved performance.

**Weaknesses:**

I have two main issues with the paper: presentation and evaluation.

In general, the paper is bit a difficult to follow. I had an idea what the paper was trying to propose, but it wasn't until I reached the dataset section that the problem, next word(s) completion, became clear. The intro reads more like a conclusion, as it uses concepts that aren't defined until later. One example: the word memory is used dozens of times before it's defined in section 3.2, and I what memory was in my head didn't quite match the details in that section. Additionally, it seems the goal (enhancing text prediction) should be a larger focus on the intro.

The figures and algorithm need more surrounding text to gently introduce a reader: the captions (of all figures too) should be substantially filled out to explain what is in the figure. For figure 1, I think a general workflow (step 1. user inputs this text, step 2. the augmentation coordinator ....) needs to be added, as the current figure is useful only if you already understand the methods.

The evaluation suffers from an important caveat: OPT is trained on wikipedia and thePile. This issue is not a showstopper for me, but it does cast some doubts on the generalizability of the method. I realize that LLMs are now being trained on the literal entire internet, so finding unseen datasets is non-trivial. I do agree, however, with the conclusions of the paper that HybridRAG is better than the baselines.


Minor:
* section 3.3, it would be nice if the full training dataset was explicitly defined as a variable
* training/test sets should be defined for the evaluation

**Questions:**

None